

# Microenvironment-related gene TNFSF13B predicts poor prognosis in kidney renal clear cell carcinoma

Mingzhe Jiang[1,*], Jiaxing Lin[1,*], Haotian Xing[1], Jun An[1], Jieping Yang[1], Biao Wang[2], Meng Yu[3] and Yuyan Zhu[1]

[1] Department of Urology, The First Hospital of China Medical University, Shenyang, Liaoning, China
[2] Department of Biochemistry and Molecular Biology, School of Life Sciences, China Medical University, Shenyang, Liaoning, China
[3] Key Laboratory of Transgenetic Animal Research, Liaoning Province, Department of Laboratory Animal Science, China Medical University, Shenyang, Liaoning, China
* These authors contributed equally to this work.

## ABSTRACT

**Background:** Kidney renal clear cell carcinoma (KIRC) affects the genitourinary system. Although treatment of KIRC in early stages can be highly successful, this type of cancer is difficult to detect until later stages of disease that are less easily treatable. Previous studies have focused on tumor cells, but have ignored the contributions of the tumor microenvironment.

**Methods:** We analyzed KIRC gene expression data from The Cancer Genome Atlas with the ESTIMATE algorithm to identify differentially expressed genes. Through protein–protein interaction network analysis, we identified clusters and picked genes from the clusters for further analysis. Differential expression, Kaplan–Meier, and univariate Cox analyses were used to select prognostic biomarkers. Gene Set Enrichment Analysis (GSEA) and Tumor Immune Estimation Resource (TIMER) analysis were used to explore the immune characteristics of these genes as biomarkers.

**Results:** Through the ESTIMATE algorithm and other system biology tools, TNFSF13B was identified as a prognostic biomarker. TNFSF13B is highly expressed in tumors, and high expression of TNFSF13B leads to poor prognosis. Further GSEA and TIMER analysis revealed that the expression of TNFSF13B was related to the immune signaling pathway and lymphocyte infiltration. Our findings strongly suggest that TNFSF13B may be a potential biomarker or target related to the tumor microenvironment for KIRC.

Corresponding author
Yuyan Zhu, yyzhu@cmu.edu.cn

## INTRODUCTION

Kidney renal clear cell carcinoma (KIRC) is a highly prevalent urinary system cancer, accounting for 80–90% of lesions in the renal space and about 3% of all new cases of tumors worldwide. About 100,000 patients die of renal cell carcinoma every year

(*Siegel, Miller & Jemal, 2017*), and KIRC is the main pathological type of this carcinoma. Although biotherapy and surgical treatment can be effective in early stages, renal cell carcinoma typically is not detected until later stages. Additionally, KIRC typically progresses rapidly and exhibits early metastasis, with overall poor patient prognosis. Therefore, identification of KIRC-related genes can provide insight into the mechanism of renal cell carcinoma and facilitate early diagnosis and treatment of renal cell carcinoma (*Cairns, 2010*).

To correlate tumor genetic composition with clinical prognosis, extensive databases of genome-scale gene expression data have been established. The Cancer Genome Atlas (TCGA, https://portal.gdc.cancer.gov/) is a public available repository of cancer genomic data, with data relevant to several important cancers. These data can be analyzed to detect genomic changes that can provide insight into cancer development. The expression patterns of different genes can be compared to clinical diagnostic criteria and may help to identify markers that are relevant to cancer diagnosis, may facilitate selection of effective treatment options, and can improve prognosis (*Krishnan et al., 2016*).

Previous studies have shown that tumor cell intrinsic genes determine the occurrence, development and evolution of KIRC (*Hsieh et al., 2017*). However, the tumor microenvironment generally has a serious impact on gene expression in cancer, as well as in KIRC and other solid tumor tissues, which can affect clinical consequences (*Chevrier et al., 2017*). The tumor microenvironment is complex, and includes inflammatory cytokines and mediators, extracellular matrix (ECM) molecules and immune, mesenchymal and endothelial cells (*Hanahan & Coussens, 2012*).

Of these components, immune and stromal cells are likely the key contributors to the tumor microenvironment, and understanding how these cells affect tumor development is essential for tumor diagnosis and assessment of prognosis. "Estimation of STromal and Immune cells in MAlignant Tumours using Expression data" (ESTIMATE) is a method that uses gene expression signatures to infer the level of non-tumor cells in tumor samples (*Yoshihara et al., 2013*). The algorithm is based on single sample gene set enrichment analysis, and the levels of immune cells and stromal cells are inferred by calculating immune score and matrix score. The ESTIMATE scores reflect the purity of tumor cells, where the higher the ESTIMATE scores, the lower the tumor purity. Previous studies have effectively used the ESTIMATE algorithm to analyze large datasets to provide insight into melanoma (*Yang et al., 2019*), breast cancer (*Chen et al., 2019*) and bladder cancer (*Wang et al., 2019*). However, despite numerous studies showing the efficacy of this data-based algorithm and the ability to calculate immune and stromal scores for KIRC, no detailed investigation has been conducted.

In this first report using the TCGA database to assess the KIRC cohort using the ESTIMATE algorithm, we identified the gene related to the microenvironment that should be further characterized for its ability to predict the prognosis of patients with KIRC.

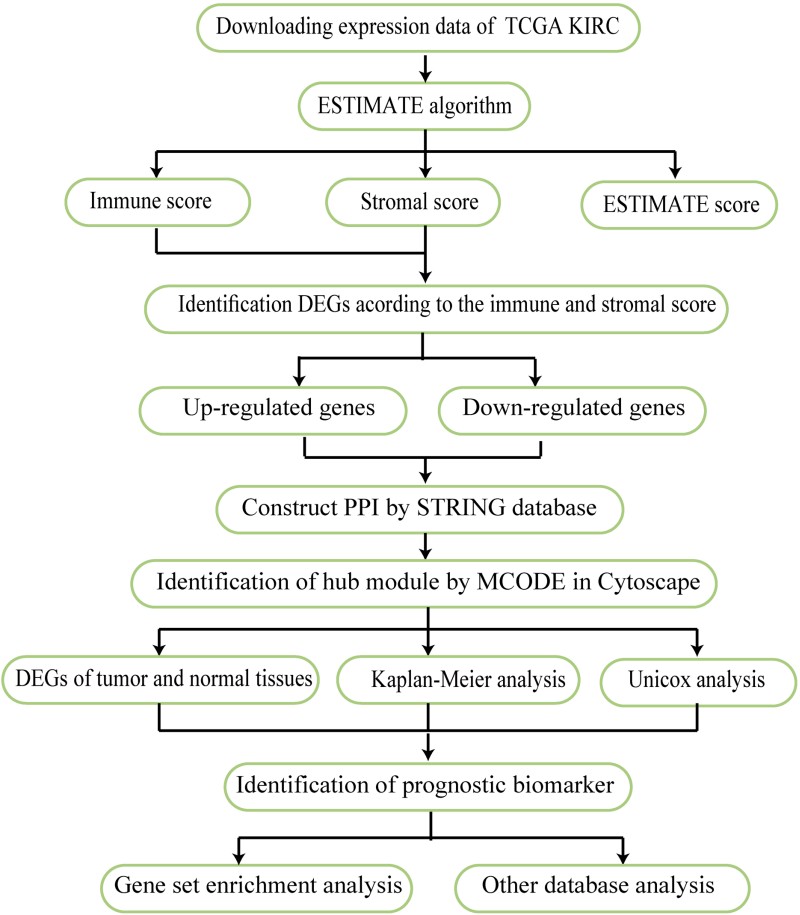

**Figure 1 Work flow used in this analysis.** The flow chart reflects the research idea of this study.

## MATERIALS AND METHODS

### Database

The work flow of this study is shown in Fig. 1. The expression profiles FPKM (level 3) and clinical data were obtained for KIRC patients using the TCGA website in March 1, 2019 (https://portal.gdc.cancer.gov/). The immune, stromal, and ESTIMATE scores of each sample were determined according to the ESTIMATE algorithm. This tool uses expression data to estimate the contributions of stromal and immune cells in malignant tumor tissue.

### Identification of differentially expressed gene

The package "limma" (https://bioconductor.org/packages/limma/, version=3.42.2) was used for expression data analysis (Ritchie et al., 2015). The input data are the expression data of KIRC. According to the median value of the scores, the patients were divided into two groups for difference analysis. We then used Wilcoxon test to test the difference in the expression of each gene between the two groups. |Log FC| > 1 (FC: fold change) and false discovery rate (FDR) $p < 0.05$ were set as the cutoff values to screen DEG.

### Heatmap and cluster analysis

The R package "pheatmap" (https://CRAN.R-project.org/package=pheatmap, version=1.0.12) was used to generate a heatmap and perform cluster analysis. The gene expression profile data were input, the gene expression heat map was drawn, and the genes were clustered according to the sample groups.

### Gene ontology and Kyoto Encyclopedia of Genes and Genomes pathway analysis

The GO and KEGG analyses were performed using the R package "clusterProfiler" (https://bioconductor.org/packages/clusterProfiler/, version=3.14.3) (*Yu et al., 2012*). FDR $p$ value < 0.05 was used to distinguish significantly enriched terms. Enter the genes, and used the functions "enrichGO" and "enrichKEGG" for enrichment analysis. The results of enrichment analysis are shown by bubble diagram.

### Construction of protein–protein interaction network and identification of functional clusters

The PPI network was constructed using the Search Tool for the Retrieval of Interacting Genes (STRING; https://string-db.org/) (*Szklarczyk et al., 2019*). Unconnected genes were removed and the confidence was set to 0.4 to obtain the PPI network. We further analyzed the PPI network with Cytoscape (Cytoscape_v3.6.1) (*Shannon et al., 2003*) and then Molecular Complex Detection (MCODE) was used to identify functional clusters (*Bader & Hogue, 2003*). The default parameters were set as follows: the degree cutoff value was 2, the node score was 0.2, the k-score was 2, and the maximum depth was 100.

### Kaplan–Meier survival analysis

Kaplan–Meier curve has time as the horizontal axis and survival rate as the vertical axis. The R packages "survival" (https://CRAN.R-project.org/package=survival, version=3.1-8) and "survminer" (https://CRAN.R-project.org/package=survminer, version=0.4.6) were applied for survival analysis. A value of $p < 0.05$ means it is statistically significant. The input matrix contained the sample's survival time, survival status, and gene expression data. Samples were grouped using a median or optimal threshold for Kaplan–Meier regression analysis.

### Univariate and multivariate cox regression analysis

Cox analysis is a semi-parametric model of survival analysis, which is used to identify the risk factors of disease and to analyze prognosis. Univariate and multivariate Cox analysis were performed with the R package "survival". The input data was the same as the Kaplan–Meier analysis, which was calculated using the function "cox" in package. The results of the analysis can be displayed in a forest map.

### Identification of practical prognostic biomarker

Three methods were used to select practical prognostic biomarker. The first was to find out that the DEGs between tumor and normal tissue, the threshold was set to |Log FC| > 1, FDR-$p$ < 0.05; the second was to use univariate Cox analysis to get the prognostic genes

($p < 0.05$); the third was to use Kaplan–Meier analysis to find out the prognostic genes ($p < 0.05$). Finally, the intersection of the three methods results was taken, and the Venn diagram is drawn with a Web tool to show (http://bioinformatics.psb.ugent.be/webtools/Venn/). We used DEG, and univariate COX here to screen prognostic markers for KIRC, and we did so because this is the most common way to obtain prognostic markers. We added Kaplan–Meier analysis because it can intuitively evaluate the prognosis of patients and has practical significance for clinical use. With the intersection of the three meaningful results, practical prognostic markers can be obtained.

## Gene set enrichment analysis

Gene Set Enrichment Analysis (GSEA) is a computational method that determines whether a group of basically defined genes exhibit statistically significant differences between two biological states (*Subramanian et al., 2005*). In this study, according to the gene expression data, the samples were divided into two groups for GSEA. One thousand genome replacement tests were performed per analysis in GSEA_4.0.1 and the expression level of each individual gene was used as a phenotypic tag. Values of $p$-value < 0.05 and FDR $p$-value < 0.05 were applied to define the pathways with statistical significance. Finally, we used the R package "ggplot2" (https://CRAN.R-project.org/package=ggplot2, version=3.2.1) to draw multiple pathways for display, the input file is the result of GSEA's analysis.

## Exploration of immune characteristics and drug targets

The R package "ggstatsplot" (https://CRAN.R-project.org/package=ggstatsplot, version=0.4.0) was used to analyze the correlation between genes expression and the scores. We used the Tumor Immune Estimation Resource (TIMER; https://cistrome.shinyapps.io/timer/) (*Li et al., 2017*) to explore the relationship between genes and immune cells infiltration. Finally, we queried the Tumor Immune System Interactions Database (TISIDB; http://cis.hku.hk/TISIDB) (*Ru et al., 2019*) for drugs that can act on the genes, using the drug information in the database from Drugbank database (https://www.drugbank.ca).

## Statistical analysis

All statistical analyses, including the operation of the ESTIMATE algorithm and gene identification, were run using R software (R x64 3.5.1). All R packages were obtained from CRAN (https://cran.r-project.org) and Bioconductor (http://www.bioconductor.org). The two groups were compared by the Wilcoxon test, and comparison between multiple groups was performed by Kruskal–Wallis test. Statistical significance was defined as $p < 0.05$.

# RESULTS

## Scores of the ESTIMATE algorithm are related to the survival and prognosis of kidney renal clear cell carcinoma

The gene expression profiles and clinical data were obtained for all KIRC patients available in the TCGA database. Using the ESTIMATE algorithm, the stromal score range was

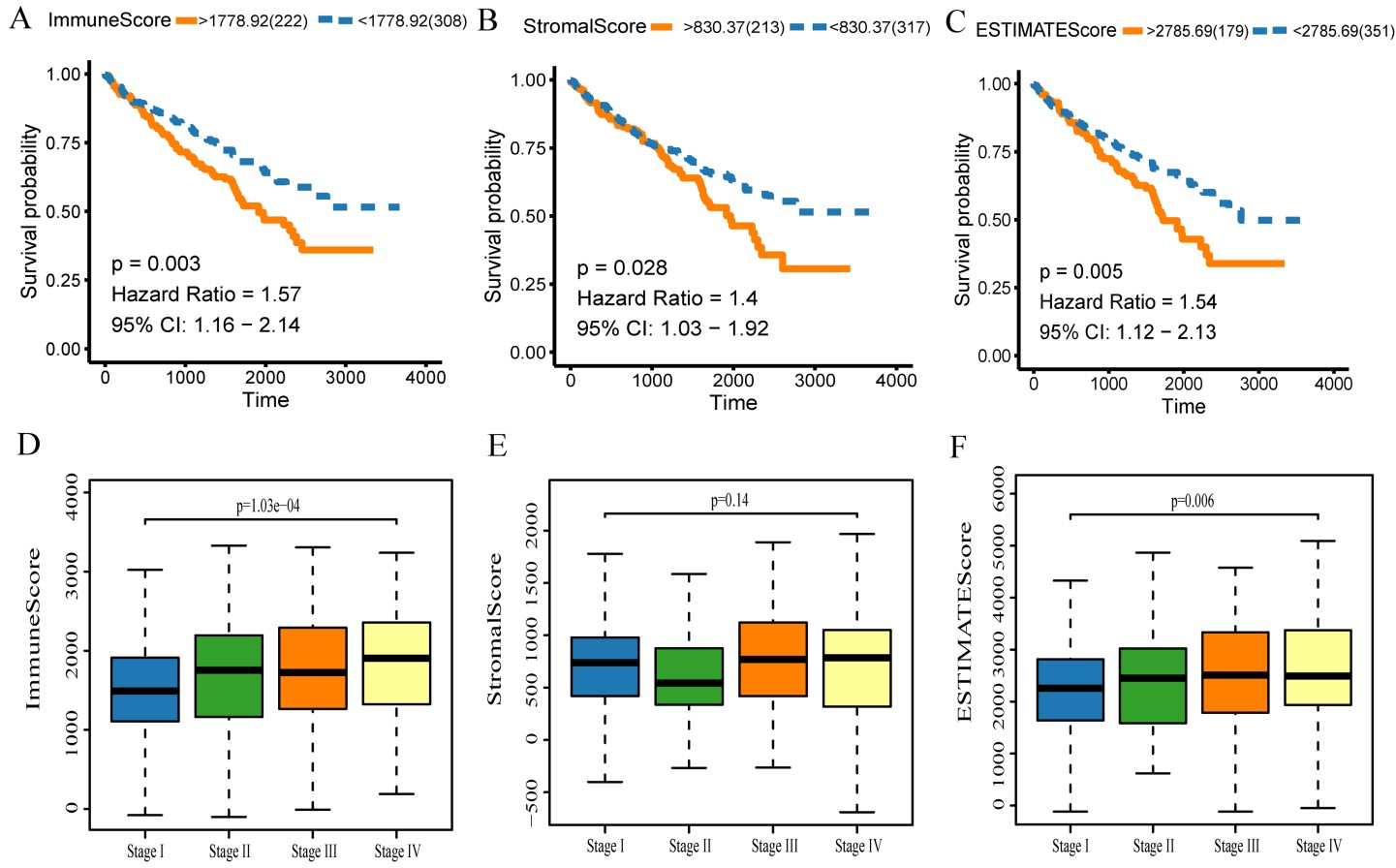

**Figure 2 The scores obtained from the ESTIMATE algorithm were related to the KIRC overall survival rate and stage.** (A) Kaplan–Meier analysis of immune score. A value of $p < 0.05$ means it is statistically significant. CI, confidence interval (B) Stromal score. (C) ESTIMATE score. (D) Box diagram of immune score in different stages. Kruskal–Wallis test was used for comparison among groups. A value of $p < 0.05$ indicates that it is statistically significant. (E) Stromal score and (F) ESTIMATE score.           

−1,433.77–1,967.19, the immune score range was −693.96–3,328.21, and the ESTIMATE score range was −2,127.72–5,091.59. To make the Kaplan–Meier analysis more statistically significant, the function "res.cut" in R package "survminer" was used to find the best segmentation point to group the samples. The scores of each group were grouped according to the optimal segmentation point calculated by the function. The best segmentation point based on immune score was 1,778.92. Relative to this point, the samples were divided into high and low immune score groups for Kaplan–Meier overall survival analysis. The results showed that the prognosis of the high immune score group was worse than the overall group ($p = 0.003$, Fig. 2A). In a similar analysis, the best segmentation points for stromal and ESTIMATE scores were 830.37 and 2,785.69, respectively. Kaplan–Meier analysis showed worse prognosis of the high score group for both stromal and ESTIMATE score-based analysis ($p < 0.05$, Figs. 2B–3C). This shows that all three scores predict the overall survival rate of patients. We generated a boxplot to show the relationships between the three scores and pathological stages of KIRC, and then analyzed the statistical significance by Kruskal–Wallis test. The results showed that the

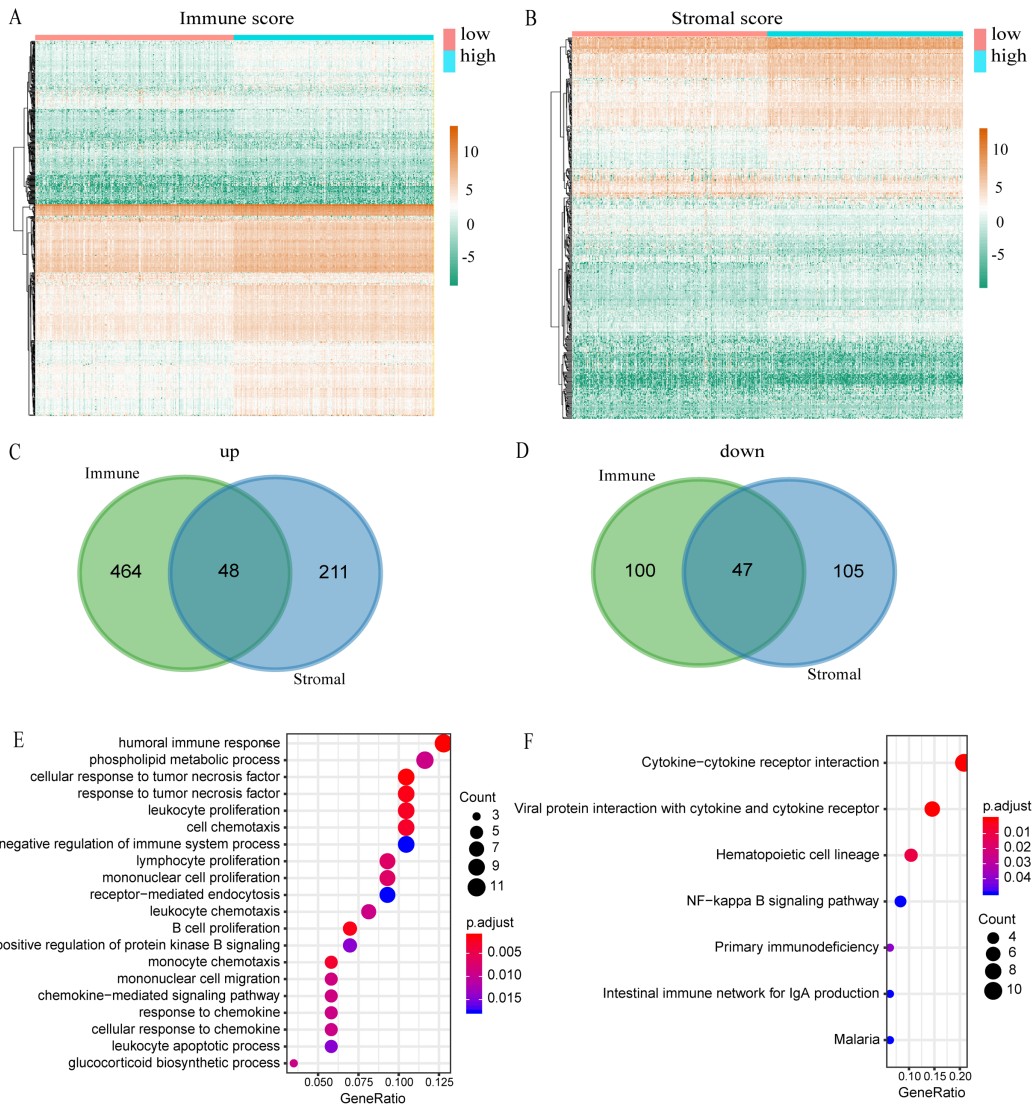

**Figure 3 Comparison of expression profiles for genes with low or high immune and stromal scores.** (A) The heatmap of DEGs between high and low immune scores. (B) The heatmap of DEGs between high and low stromal scores. (C) Venn diagram shows the number of up-regulated DEGs. (D) Venn diagram shows the number of down-regulated DEGs. (E) A bubble chart shows the top 20 enriched GO terms. The GO analysis had an FDR of less than 0.05. (F) The KEGG enrichment analysis exhibited an FDR of the six pathways that were less than 0.05.

immune scores (Fig. 2D, $p < 0.001$) and ESTIMATE scores (Fig. 2F, $p = 0.006$) were both significant in different pathological stages, and increased with the increase of pathological stages. However, there were no differences in the stromal scores among different stages (Fig. 2E, $p = 0.140$).

## Mining differential genes in gene expression profiles according to immune score and stromal score

A total of 539 KIRC samples were first grouped according to the median of immune score and then a heatmap of gene expression was constructed (Fig. 3A). In the differential

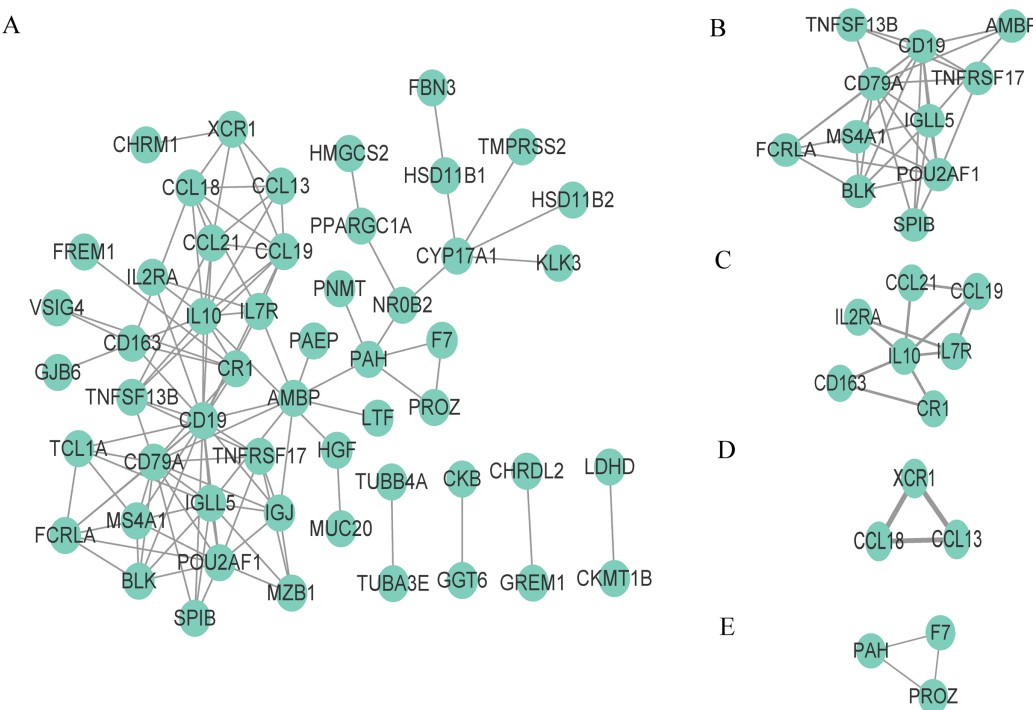

**Figure 4 Construction of PPI network and identification of clusters.** (A) The PPI network, (B) Cluster 1, (C) Cluster 2, (D) Cluster 3 and (E) Cluster 4.

gene analysis of immune score, 512 genes were up-regulated and 147 genes were down-regulated ($|\log FC| > 1$, $p < 0.05$). Similarly, the samples were grouped according to the median of stromal score and constructing heat maps of gene expression (Fig. 3B). A total of 259 genes were up-regulated and 152 genes were down-regulated in the groups with high and low stromal scores ($|\log FC| > 1$, $p < 0.05$). Since both immune and stromal factors are part of the immune microenvironment, we took the intersection between immune up-regulated genes and stromal up-regulated genes. Then we also took the intersection between immune down-regulated genes and stromal down-regulated genes. The Venn map shows that 48 genes are generally up-regulated (Fig. 3C) and 47 genes generally are down-regulated (Fig. 3D). We decided to focus on these 95 DEGs, which we propose are genes that are important to the microenvironment. We investigated the function of these genes by GO and KEGG enrichment analysis. The most significant term of GO enrichment analysis is Humoral immune response, while that of KEGG enrichment analysis is Cytokine–cytokine receptor interaction. These results suggest that these genes are related to immunity (Figs. 3E and 3F).

## Construction of PPI network and identification of functional clusters

To better understand the interaction of DEGs in the microenvironment, we used the STRING tool to construct a PPI network for the 95 DEGs and visualize the network with Cytoscape (Fig. 4A). The network consists of 53 nodes and 110 edges, and 42 genes are not displayed in the network because they are not linked to the other genes. We then used the tool MCODE of Cytoscape to identify the functional clusters; this tool aggregates

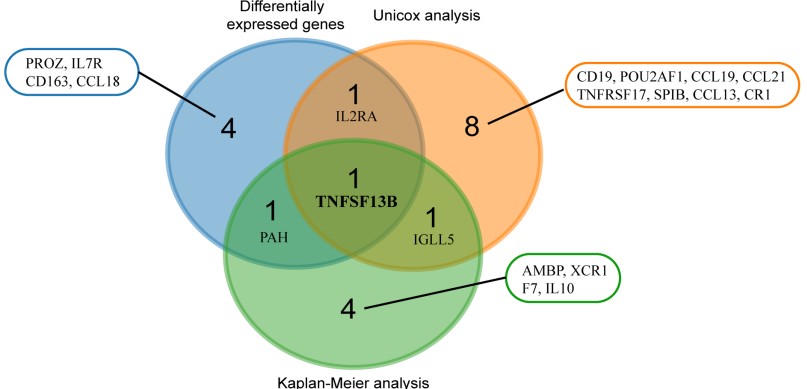

**Figure 5 Venn analysis of results from three analyses.** Blue circles indicate differentially expressed genes of the tumor and normal tissue ($|logFC > 1|$, $p < 0.05$). Red circles indicate genes with significant results from univariate analysis ($p < 0.05$). Green circles mean genes with significant results from Kaplan–Meier analysis ($p < 0.05$).

highly related genes into a cluster. A total of four clusters which contain 24 genes were identified (Figs. 4B–4F). We took the 24 genes for the next step of analysis.

## Identification of KIRC practical prognostic biomarker

We used three methods to screen practical prognostic biomarkers from the 24 genes. The expression profiles of 24 genes were analyzed to find the DEGs of tumor and normal tissues. A total of seven DEGs (CD163, CCL18, TNFSF13B, IL7R, IL2RA, PROZ and PAH) were found ($|logFC| > 1$, $p < 0.05$). According to the median expression of each gene, the genes were divided into high expression group and low expression group. Kaplan–Meier analysis of the 24 genes showed that seven genes (TNFSF13B, XCR1, IGLL5, PAH, IL10, F7 and AMBP) showed significant differences in expression ($p < 0.05$). Univariate Cox analysis of the 24 genes showed that expression levels of 11 genes (CCL13, IL2RA, TNFSF13B, IGLL5, CCL19, SPIB, CCL21, CR1, TNFRSF17, CD19, POU2AF1) were significantly different ($p < 0.05$). The results of the three analyses were combined in a Venn diagram (Fig. 5). TNFSF13B is the only gene found in all three analysis.

## High expression of TNFSF13B leads to poor prognosis of KIRC

The expression of TNFSF13B in tumor tissue is higher than that in normal tissue ($p < 0.001$, Fig. 6A). High expression of TNFSF13B leads to poor prognosis of KIRC ($p = 00019$, Fig. 6B). The clinical factors and TNFSF13B were analyzed by univariate and multivariate regression analysis. Univariate Cox analysis showed that Age, Grade, Stage, T (T stage), N (Lymph node), M (Tumor metastasis), and TNFSF13B ($p < 0.001$) were prognostic factors (Fig. 6C), it has very significant statistical significance. Multivariate Cox analysis showed that Grade and Age were independent prognostic factors (Fig. 6D). The result indicated that TNFSF13B was a prognostic factor, but it could not analyze the prognosis independently of other clinical factors. The relationship between gene expression and stage/grade was analyzed by box diagram, and Kruskal-Wallis test was used for comparison among groups. The results showed that the TNFSF13B expression was

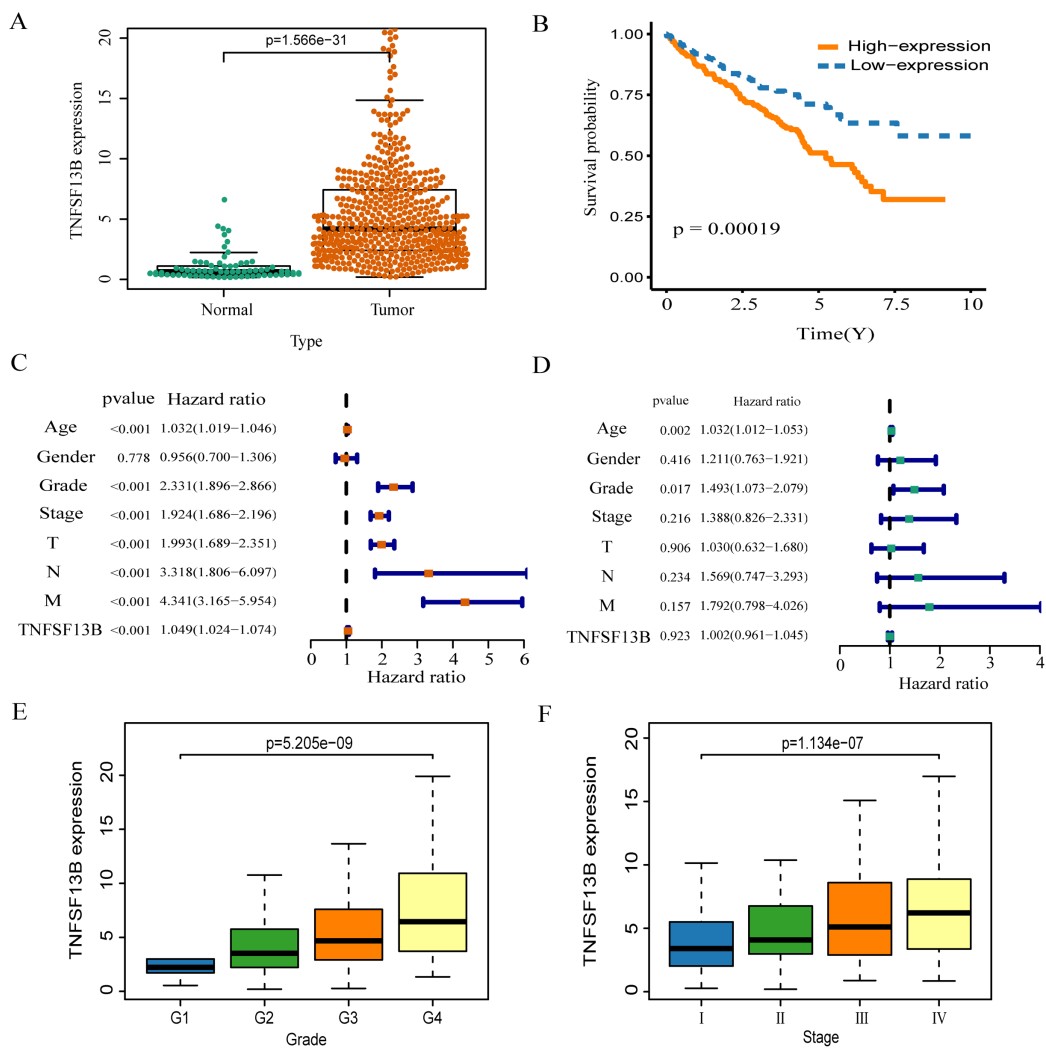

**Figure 6 Analysis of the relationship between TNFS13B and clinical factors.** (A) Scatter plot of TNFS13B expression in tumor and normal tissues. Wilcoxon test was used to compare the two groups. (B) Kaplan–Meier analysis of TNFS13B. (C) Forest plot of univariate COX regression analysis for clinical factors. (D) Forest plot of multivariate COX regression for clinical factors. (E and F) Box diagram of TNFSF13B expression under different grades and stages. Kruskal–Wallis test was used for comparison among groups. A value of $p < 0.05$ indicates that it is statistically significant.

different in different stages and grades ($p < 0.001$, Figs. 6E and 6F), and the TNFSF13B expression increased with the increase of stage and grade.

## There is a positive correlation between TNFSF13B and tumor microenvironment.

By analyzing the correlation between immune/stromal/ESTIMATE score and TNFSF13B expression, it was found that TNFSF13B was positively correlated with immune ($R = 0.67$, $p < 0.001$, Fig. 7A), stromal ($R = 0.43$, $p < 0.001$, Fig. 7B), and ESTIMATE ($R = 0.64$, $p < 0.001$, Fig. 7C). We next queried the relationship between TNFSF13B and immune cell infiltration in the TIMER database (Figs. 7D–7J), and found that its expression was

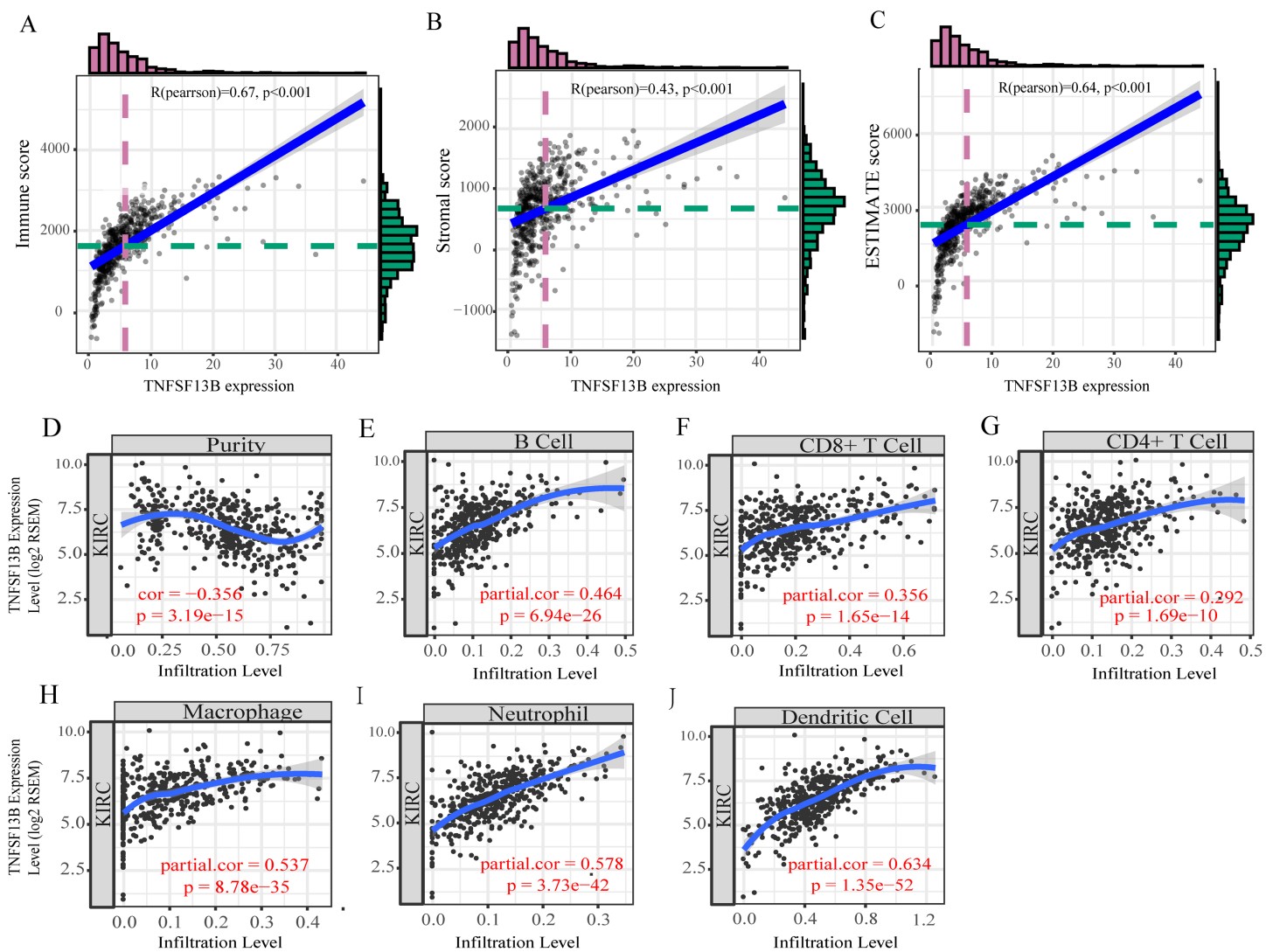

**Figure 7 Tumor microenvironment exploration of TNFSF13B.** (A) Correlation scatters diagram of the immune score and TNFSF13B expression. (B) Stromal score. (C) ESTIMATE score. (D) Correlation scatters Diagram of Tumor purity and TNFSF13B expression. (E) B cell. (F) CD8+ T cell. (G) CD4+ T cell. (H) Macrophage. (I) Neutrophil and (J) Dendritic cell. A value of $p < 0.05$ indicates that it is statistically significant.

positively correlated with tumor purity and positively correlated with immune cells. TNFSF13B expression value showed maximum partial spearmanm's correlation with dendritic cells (partial cor = 0.634, $p$ =1.35E−52).

## Gene set enrichment analysis of TNFSF13B

GSEA is a powerful analytical method used to interpret gene expression data. We sequentially analyzed gene enrichment and performed GSEA for TNFSF13B. We used the Molecular Signatures Database (MSigDB, http://www.broadinstitute.org/msigdb) of Collection (c2.cp.kegg.v7.0.symbols.gmt) (*Liberzon et al., 2011*, *2015*). We set the threshold for FDR $p$-value < 0.05 and $p$-value < 0.05. Significantly enriched signal transduction pathways were identified based on normalized enrichment score (NES). We identified

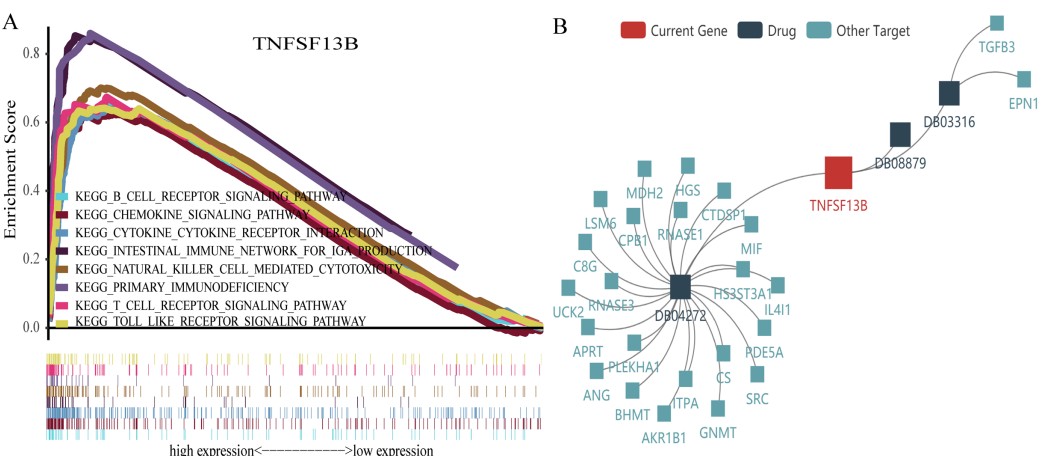

**Figure 8  Gene set enrichment analysis and drug target network.** (A) A broken line above represents an enrichment pathway, and the straight line below represents every gene in the pathway and (B) network diagram of drugs and acting genes.

pathways with high NES that are related to immune function. Because few meaningful signal pathways were enriched by low gene expression, we only present the signal pathways enriched for high gene expression (Fig. 8A). There was clear enrichment in immune-related signaling pathways. Finally, we found three drugs that have effects on TNFSF13B in the TISIDB database (Fig. 8B), 1, 4-dimethyl Dioxide (DB03316), Citric Acid (DB04272), and Belimumab (DB08879). Among them, Belimumab is an intravenous immunosuppressant and the only target for the adjuvant treatment of systemic lupus erythematosus (SLE) (*Scott, Burness & McCormack, 2012*).

## DISCUSSION

Previous experiments of KIRC focused on animal tumor models, in vitro tumor cell lines, or patient tumor samples (*Capitanio & Montorsi, 2016*). However, the KIRC microenvironment must also be considered. With increased exploitation of whole genome sequencing, TCGA has been established as an extensive tumor database for free use by the research community. These resources facilitate data analysis of many cancers, including KIRC (*Krishnan et al., 2016*).

The goal of this study was to identify genes that are related to the microenvironment and overall survival of KIRC. Gene expression patterns were compared for cases with high and low immune scores, and 95 genes were identified as potentially involved in functions related to the ECM or to immune response. To assess prognostic ability, we analyzed 95 differentially expressed genes from groups with high immune score (or high stromal score), and used GO and KEGG enrichment analysis to determine that many of these genes are components of the tumor microenvironment. Next, we constructed a protein–protein interaction network of these 95 genes, and identified four clusters which contained 24 genes. We performed differential expression, Kaplan–Meier, and univariate Cox analyses for the 24 genes. The TNFRSF13B gene was identified in all three analyses. TNFRSF13B was overexpressed in tumors, Kaplan–Meier analysis showed

that high expression of TNFRSF13B led to poor prognosis, and univariate Cox showed that TNFRSF13B was a risk factor. The expression of TNFRSF13B is positively correlated with tumor stage and grade. These results suggest that TNFRSF13B is a potential prognostic marker.

TNFRSF13B is an essential factor in the tumor microenvironment of KIRC. There was a high positive correlation between TNFRSF13B and ESTIMATE score, indicating that the more TNFRSF13B expression, the less the content of tumor cells, the more the content of immune cells and stromal cells. Renal cell carcinoma has a good effect on immunotherapy (29120911), so we inquired about the relationship between TNFRSF13B and six kinds of immune cells. TNFRSF13B has a high correlation with immune cells. GSEA results showed that TNFRSF13B was involved in many immune-related pathways, including the T cell receptor signaling pathway and so on. TNFSF13B belongs to the tumor necrosis factor (TNF) ligand family, and is also named BAFF. TNFSF13B variants are associated with multiple sclerosis and SLE (*Steri et al., 2017*), and TNFSF13B is a key factor in the survival of B lymphocytes (*Cremasco et al., 2014*). Previous study suggested a potential role of these TNFSF members including TNFRSF13B in renal tumor biology (*Pelekanou et al., 2011*). Recently, it has been reported that TNFSF13B is related to the prognosis and immune infiltration of KIRC (*Li et al., 2019*). This article starts directly from TNFSF13B to explore its prognosis and immune infiltration. However, we started from the tumor microenvironment, went through many aspects of screening, and finally selected the representative gene TNFSF13B. Their research and our exploration have jointly confirmed the important role of TNFSF13B in KIRC. Through our research, TNFSF13B can not only predict the prognosis of KIRC, but also be an important medium for KIRC to transmit information between the tumor cell and tumor microenvironment, and is a potential therapeutic target.

The interaction between KIRC and the tumor microenvironment seriously affects the progress and evolution of this carcinoma, and subsequently affects the recurrence and overall prognosis of patients. Previous reports have provided a detailed analysis of how the activation of tumor intrinsic genes shapes the tumor microenvironment. In our current work, we analyzed the genetic characteristics of the microenvironment that affect the development of KIRC. Our results may help explain the complex interactions between the tumor and the tumor environment in KIRC. Of course, our research also has some shortcomings. In the future, we need more data to prove the prognostic ability of TNFSF13B and also need experiments to prove the relationship between TNFSF13B and microenvironment.

## CONCLUSIONS

We analyzed KIRC gene expression data from The Cancer Genome Atlas with the ESTIMATE algorithm to identify the differentially expressed genes. Through multiple analyses, we identified TNFSF13B as a potential biomarker and target for KIRC. Further study of TNFSF13B may improve understanding of the potential relationships between the tumor microenvironment and the prognosis of KIRC.

## LIST OF ABBREVIATIONS

| | |
|---|---|
| **KIRC** | Kidney renal clear cell carcinoma |
| **TCGA** | The Cancer Genome Atlas |
| **ESTIMATE** | using expression data to estimate stromal and immune cells in malignant tumor tissue |
| **DEGs** | differentially expressed genes |
| **FC** | fold change |
| **GSEA** | Gene Set Enrichment Analysis |
| **FDR** | false discovery rate |
| **PPI** | protein–protein interaction, |
| **TIMER** | Tumor Immune Estimation Resource |

## ACKNOWLEDGEMENTS

We appreciate the data obtained from the TCGA database in this study.

### Funding

This work was supported by the National Natural Science Foundation of China (81672523, 81472404, 81472403 and 81572831), the 2018 Support Plan for innovative talents in Colleges and Universities of Liaoning Province, the 2018 "Million Talents Project" funded by the Project of Liaoning Province, the 2019 Key R&D projects of Shenyang, and the Science and Technology Research Project of the Education Department of Liaoning Province (LK201616). The funders had no role in study design, data collection and analysis, decision to publish, or preparation of the manuscript.

### Grant Disclosures

The following grant information was disclosed by the authors:
National Natural Science Foundation of China: 81672523, 81472404, 81472403 and 81572831.
Project of Liaoning Province.
Key R&D projects of Shenyang.
Science and Technology Research Project of the Education Department of Liaoning Province: LK201616.

### Competing Interests

The authors declare that they have no competing interests.

### Author Contributions

- Mingzhe Jiang conceived and designed the experiments, performed the experiments, analyzed the data, prepared figures and/or tables, and approved the final draft.
- Jiaxing Lin conceived and designed the experiments, performed the experiments, analyzed the data, authored or reviewed drafts of the paper, and approved the final draft.

- Haotian Xing performed the experiments, prepared figures and/or tables, and approved the final draft.
- Jun An performed the experiments, prepared figures and/or tables, and approved the final draft.
- Jieping Yang performed the experiments, prepared figures and/or tables, and approved the final draft.
- Biao Wang analyzed the data, authored or reviewed drafts of the paper, and approved the final draft.
- Meng Yu analyzed the data, authored or reviewed drafts of the paper, and approved the final draft.
- Yuyan Zhu conceived and designed the experiments, authored or reviewed drafts of the paper, and approved the final draft.

## Data Availability

The raw data are available in the Supplemental File.

## Supplemental Information

Supplemental information for this article can be found online at http://dx.doi.org/10.7717/peerj.9453#supplemental-information.

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
