# Peer review of "Microenvironment-related gene TNFSF13B predicts poor prognosis in kidney renal clear cell carcinoma"

_PeerJ, doi:10.7717/peerj.9453_

## Round 0.1 · original submission · Major Revisions

To identify early stage diagnostic markers by considering the tumor microenvironment, this study performs a network-based analysis of RNA-sequencing data of KIRC patients.

As the reviewers highlighted, in its current form, the manuscript is difficult to follow and many of the analyses seem arbitrary. There is not enough explanation regarding the followed methodologies. The reviewers asked several relevant questions regarding the methodology but one basic question is that "How are hub genes found?" and "How is TNFSF13B selected as the hub gene from previously declared six “hub genes” or among other previously mentioned 23 hub genes?" The statistical aspect of this manuscript is not explained sufficiently, as required for publication.

Also, the English language should be improved to ensure that an international audience can follow the text. There are some typos as noted by the reviewers. And some confusing points regarding the abbreviations, as listed in reviewer's comments.

Reviewer 1 ·

Basic reporting

The written English is readable and satisfactory. There are some typos in the manuscript (line 22, 113 etc.).
Literature references are sufficient in terms of background coverage and biological validation. However, the citation style in the text is confusing. For example, in line 53, there are four references cited by only using first letter of authors, this mistake is repeated in whole manuscript.
Resolution of some figures is poor, e.g., Figure 3 (all fonts in small captions are unreadable), Figure 7B.

Experimental design

The manuscript did not develop a novel algorithm for RNA-sequencing analysis or for biological network analysis, so it might not be considered as a novel research paper.
This study could be interpreted as a case-study. The found biomarker might initiate new in-vivo / vitro studies for KIRC treatment. However, this fact does not change being a case-study type of manuscript.

The method section needs improvements in terms of following questions and suggestions:
1. How were down-regulated genes selected by using only LogFC > 1 cut-off?
2. In line 100-101, "the number of connections of gene node" should be expressed as "degree". This definition should be reflected to the rest of the manuscript.
3. In line 102, identification of a hub node in a PPI network is bounded by nodes with a higher than 5 connections (degree). How was this cutoff "5" chosen?
4. The details of ESTIMATE algorithm can be explained in the beginning of the method section. Although it is used as a tool, it could be helpful to give more technical details to replicate this study.

Validity of the findings

There are some questions about explanations of results in this study:
1. In line 143, authors wrote that "the KIRC samples were divided into high score group and low score group". What are the numerical ranges of these high and low scores? How did they set these score values?
2. In line 172, the construct PPI sub-network covers 53 nodes and 110 edges. why did it not cover 95 of DEGs identified in the previous step?

Additional comments

The study performs a network-based analysis of KIRC patient RNA-sequencing samples to identify some early stage diagnostic markers by considering the tumor microenvironment. Verification of identified markers on the independent datasets is remarkable analysis of the study.

Authors did not develop a novel analysis algorithm for RNA-sequencing or biological network analysis, they applied well-known algorithms and online data sources. So, this study could be interpreted as a case-study. The identified biomarker, TNFSF13B, was analyzed in other cancers, but not mainly kidney one. This finding could have contribution to initiate other experimental studies for KIRC treatment.

This manuscript can be evaluated as new research paper, when it performs analysis for another cancer type by applying the same computational pipeline and considering effects of the tumor microenvironment.

·

Basic reporting

1. No need to mention the abbreviations of terms that are mentioned only once in the abstract.
2. Most references were cited using first name (or an abbreviation), year for example line 40, 46 and 53. Please revise. References to data or tools should be cited on the first mention for example TCGA (line 48).
3. TCGA is currently hosted by GDC. A direct link the current host would be the most helpful.
4. A brief explanation of what the ESTIMATE algorithm does might be needed. Especially what each of the scores means exactly in the context of the study.
5. The description of the differential expression analysis (line 86) is not sufficient. The section should mention input data, preprocessing, quality controls and the comparisons that were attempted.
6. References to some of the tools (R packages) were not included in the text. For example, heatmap, beeswarm, ggplot2 and R itself. If no papers/books are recommended by the creators to cite, please cite them as software.
7. On some occasions, the authors didn’t mention the specific tools that were used to conduct certain steps of the analysis. For example, calculating the node degrees (line 101) and the survival curves (line 104).
8. The heatmaps (Figure 3A&B) are not very informative. Different scaling, color schemes or clustering might be helpful in showing the trends in the expression profiles. The labels (rows and columns) are very small. Alternative graphs such as volcano plots might be considered to show the difference in expression between low and high scores.
9. The functional enrichment analysis reported in Figure 3E&F was only briefly mentioned in the text (line 69). The findings of this analysis should be described in the results or removed if not essential to the study. In addition, the x-axis labels of those two figures are missing.
10. Figures 5A-F (right) were referred to in the text as scatter maps (line 183). Those look more like a box plot with point or jitter overlayed on top. Also, the unified scale of the y-axis in those plots (0-80) makes it difficult to see the spread of the points. Rescaling or different y ranges might be useful.
11. The authors state that the cutt-off of the statistical significance in expression between low and high scores were based on log FC and (p)-value (line 161-162). Adjusting for multiple testing using something like FDR should be considered or stated if already used. Moreover, it seems like the intersection between immune and stromal score-based differentially expressed genes rather than the “interaction” (line 163) was used to select the genes of interest.

Experimental design

1. The section describing “selection of hub genes with oncomine database” is confusing. First, the specific aim of that step of the analysis is not clear. How can hub genes be selected based on a meta-analysis? Second, the datasets on which the meta-analysis was based were neither cited nor described properly. Finally, the verification eluded to (line 121) was based on the same datasets.
2. Figure 5 shows the prognostic values of six hub genes. These predictors were not shown to be independent. In addition, the relation of these factors to other clinical characteristics of the samples/patients can be shown to support or explain their values.
3. Although 95 differentially expressed genes were suggested to be of interest, only 53 are found in the protein-protein interaction network. There seems to be an unacknowledged loss of information in this step.
4. The selection of TNFSF13B as a “hub gene” from previously declared six “hub genes” is unjustified or at best unexplained.
5. Figure 6B-E show an upward trend of the centered rations of the gene of interest with the tumor stage/grade but no correlation stats were reported as claimed in the text (line 192).

Validity of the findings

This study stresses the importance of the tumor microenvironment in renal carcinoma. However, due to the lack of sufficient details of the methodology (above, basic reporting) and some arbitrary analysis decisions (above, experimental design), the presented findings doesn’t seem to justify the claims of the study. The above mentioned issues can be summarized in two points. First, some of the analysis were taken as support for the findings of the preceding steps. For example, it is not clear why a meta-analysis of gene expression in some datasets would support the importance of highly connected genes in a network that was derived from differentially expressed genes between samples with different immune or stromal scores. Second, some of the claims were not rigorously investigated. For example, the prognostic values of the six “hub genes” were not tested in a multivariate analysis and were not shown to be independent predictors. Another example was to claim the a gene of interest is correlated with tumor grade or stage without showing a correlation statistics.

Reviewer 3 ·

Basic reporting

No comment

Experimental design

Overall the research question is meaningful and the methods used to address it are sound.
However the methods section should be further clarified:
1. How are hub genes found? It seems PPI and survival analysis is used as 1 method and Oncomine database is used as another. However, it’s not clear how the information from the 2 methods is integrated. It would be helpful if details are provided about number of genes found by all methods, overlap between them and significance of overlap.
2. Version of all software should be mentioned along with the libraries/parameters used. E.g.: R version?, GSEA version?, limma version? Etc.
3. For GSEA analysis give more details. What gene sets were included/excluded. E.g. greater than 15 or less than?
4. Line 87 mentions LogFC. What is the base here?
5. Give details about ESTIMATE algorithm. How are the 3 scores calculated: immune, stromal, ESTIMATE.

Validity of the findings

Overall the authors have mined the TCGA database and gotten some interesting results for KIRC. However the manuscript in its current form is difficult to follow and many of the analysis seem arbitrary. It would greatly help if the authors could provide a rationale for some of their choices. E.g.: explain why hub genes should be identified through their method, difference between the 2 methods and the reason for taking the intersection.
Some other analysis/text that are unclear:
1. The authors wanted to determine the role of immune and stromal cells to the tumor microenvironment in KIRC. They found survival to be anti-correlated to immune score however not stromal scores. It’s important to explain over here the differences between the different scores.
2. It’s also not clear why the authors took an intersection of the differential genes found by immune & stromal scores when stromal scores show no differences between stages. These 95 genes should be provided in a table.
3. Its not clear why the authors only select TNFS13B as a hub gene in the end when earlier they had found 23 hub genes. What happened to the rest?
4. Also its confusing how hub genes seem to have multiple names in the manuscript: e.g: central node genes, hub nodes etc.

---

## Round 0.2 · Minor Revisions

I would like to thank to the authors; they took the comments seriously and updated the original manuscript with in the light of reviewer's suggestions. The English proof-reading elevated the language of manuscript. The correction of basic typos helps to increase quality. Improving the resolution of figures also helps to interpret the major results of the study. The better explanation of applied analysis definitely improved the quality of paper, the representation of results makes an easier understanding of the study.

However, reviewer 2 has some concerns. I would like the authors to answer his questions and add the necessary explanations.
Reviewer 3 asked the authors to add a reference to a recent paper and discuss the findings as compared to this new paper in a few sentences.

Reviewer 1 ·

Basic reporting

The English proof-reading elevated the language of manuscript. The correction of basic typos helps to increase quality.

Improving the resolution of figures also helps to interpret the major results of the study.

Experimental design

The suggestions made for explanation of the ESTIMATE algorithm was taken seriously and added an extra paragraph.

The MCODE algorithm was not mentioned at all in the original submission. Due to some questions about network clustering etc. authors also added the usage purpose of this algorithm into manuscript, this extension also found to be helpful, since it puts together the whole picture in analysis stage.

Validity of the findings

The statistical questions of reviewers about the interpretation of the results were generally replied and introduced necessary data and sections into the manuscript.

Additional comments

I thank to authors; they took comments seriously and updated the original manuscript with the light of several suggestions. The better explanation of applied analysis definitely improved the quality of paper, the representation of results makes an easier understanding of the study.

·

Basic reporting

no comment

Experimental design

no comment

Validity of the findings

no comment

Additional comments

Citation for some of the tools are still missing. For example, the “survival” and “survminer” R packages. Moreover, in addition to mentioning the analysis tools, a brief description of the goal, input and output of the analysis is always useful.
The reported three analyses in place of Oncomine were not fully described in the methods section. More specifically, which groups were compared in the differential expression and which genes were included in the survival and cox regression? Finally, no rationale was given as to why the intersection of seemingly very different analysis is of particular importance.
In the univariate and the multivariate analyses, TNFSF13B had a very small effect size or non-significant one. Despite being shown to predict survival in KM analysis, the cox analysis does not support this finding.
The authors claim that TNFSF13B is a potential biomarker and that it is related to the tumor microenvironment. The first point is not strongly supported by the revised analysis. The second point is only touched on briefly. The findings that many genes are differentially expressed between groups of different stromal/immune scores is not surprising. TNFSF13B is not shown to be representative or key to these differences.

Reviewer 3 ·

Basic reporting

No comment

Experimental design

The points raised by me have been addressed sufficiently.

Validity of the findings

The authors have significantly improved the manuscript and the conclusions are well supported.

Additional comments

There is a recent study published on TNFSF13B and KIRC that on a brief glance seems complementary and very close to your work. (TNFSF13B correlated with poor prognosis and immune infiltrates in KIRC: a study based on TCGA data..by Song Li, Junqing Hou, Zhen Wang, Junkai Chang, Weibo Xu). You should add that reference and discuss any agreement/disagreement in results.

---

## Round 0.3 · Minor Revisions

The article is substantially improved since its first submission. I appreciate the detailed responses from the authors. The structure of the revised manuscript is much easier to follow now. Now it gives some interesting insights about the role of TNFSF13B. There is only one minor comment by one of the reviewers. Please find this comment below regarding the citation to a software and an explanation regarding the intersection.

·

Basic reporting

no comment

Experimental design

no comment

Validity of the findings

no comment

Additional comments

The revised manuscript addresses most of the issues raised in the review. Two issues remain however.
1. If a software tool doesn’t have a directly related paper by the creator, it can be cited as a software program. Use the `citation(“<package name>”)` in R to get the recommended citation for a given package. Alternatively, many of the R packages have books or web pages that can be used as citations.
2. No rationale was given as to why the intersection of seemingly different analysis is of particular importance.

Reviewer 3 ·

Basic reporting

No comment

Experimental design

No comment

Validity of the findings

No comment

Additional comments

I want to thank the authors for incorporating all reviewer comments. The article is substantially improved and gives some interesting insights about the role of TNFSF13B.

---

## Round 0.4 · accepted · Accept

There were two requests by one of the reviewers. Both of them are addressed in the current version of the manuscript.